# Evaluating access to oral anti-diabetic medicines: A cross-sectional survey of prices, availability and affordability in Shaanxi Province, Western China

**Caijun Yang**[1,2], **Shuchen Hu**[1,2], **Yanbing Zhu**[3,4], **Wenwen Zhu**[1,2], **Zongjie Li**[1,2], **Yu Fang**[1,2]*

**1** The Department of Pharmacy Administration and Clinical Pharmacy, School of Pharmacy, Xi'an Jiaotong University, Xi'an, China, **2** The Center for Drug Safety and Policy Research, Xi'an Jiaotong University, Xi'an, China, **3** Department of Pharmacology, Health Science Center, Xi'an Jiaotong University, Xi'an, China, **4** Shaanxi Food and Drug Administration, Xi'an, China

* yufang@mail.xjtu.edu.cn

## Abstract

### Objectives

To assess the availability and affordability of oral anti-diabetic medicines in Shaanxi Province, Western China.

### Methods

In 2015, the prices and availability of 8 anti-diabetic medicines covering 31 different dosage forms and strengths were collected in six cities of Shaanxi Province. A total of 72 public hospitals and 72 private pharmacies were sampled, using a modified methodology developed by the World Health Organization (WHO) and Health Action International (HAI). Medicine prices were compared with international reference prices to obtain a median price ratio. For urban residents, affordability was assessed as the lowest-paid unskilled government workers to purchase cost of standard treatment in days' wages; for rural residents, days' net income was used.

### Results

The mean availabilities of originator brands (OBs) and generics were 34.3% and 28.7% in public hospitals, and 44.1% and 64.4% in the private pharmacies. OBs and the lowest priced generics (LPGs) were procured at 12.38 and 4.52 times the international reference price in public hospitals, and 10.26 and 2.81 times the international reference prices in private pharmacies. Treatments with OBs were unaffordable even for urban residents. The affordability of the LPGs was good, except for acarbose, repaglinide and pioglitazone.

**Data Availability Statement:** All relevant data are within the manuscript and its Supporting Information files.

**Funding:** This work was supported by National Natural Science Foundation of China (http://www. nsfc.gov.cn/) under award number 71503197 [PI CJ Yang] and number 71473192 [PI Y Fang] and"the Fundamental Research Funds for the Central Universities" (http://www.xjtu.edu.cn) without award number [PI CJ Yang].The funders had no role in study design, data collection and analysis, decisionto publish, or preparation of the manuscript.

**Competing interests:** The authors have declared that no competing interests exist.

## Conclusions

Most anti-diabetic medicines cannot met the WHO's availability target (80% availability) in Shaanxi Province. The high prices of OBs had severely influenced the affordability of medicines, especially for the rural residents. Effective policies should be initiated to ensure the Chinese people a better access to more affordable anti-diabetic medicines.

## Introduction

China had the largest diabetes epidemic in the world, and the prevalence continued to increase [1]. According to national surveys, the prevalence of diabetes in China was 0.9% in 1980, and it became 2.5% in 1994 and 2.6% in 2002 [2]. Furthermore, based on the data from the most recent national survey in 2013, the estimated prevalence of diabetes increased sharply to 10.9% [3], representing an estimated 148.3 million population with diabetes. Although an additional diagnostic criterion $HbA_{1c}$ (glycated hemoglobin$A_{1c}$, concentration of 6.5% or higher) was included after 2010, these data documented a rapid increase in diabetes in China. Diabetes have become a major public health problem for Chinese population.

The prevalence of diabetes raised a serious social economic concern. Because of the disorder of glucose metabolism, diabetes affected multiple organ systems and was associated with a wide range of vascular and nonvascular conditions [4]. Therefore, patients with diabetes tended to have huge medical expenditures and to take more medications to maintain their quality of life and longevity. One research showed that, the annual per capita medical spending for patients with diabetes was estimated to be more than twice that for patients without diabetes [5]. Health expenditure related to diabetes and its complications (such as periodontal disease, blindness, amputations, end-stage renal disease and cardiovascular disease, etc.) in China reached US$51 billion in 2015, which ranked second in worldwide health expenditure [1]. Among the total health expenditure, medication cost represented a substantial proportion, taking 32%-62% of total expenditure in low-and middle-income countries [6]. In China, for general patients, the medication cost took more than 47% of the total health expenditure for outpatients and nearly 35% for inpatients in 2016 [7]. For patients with diabetes, this number was bigger. One research about insured Type 2 diabetes patients with chronic kidney disease under hospitalization in China showed that the medication cost accounted for more than 47% of the total cost, and among which nearly 30% were anti-diabetics [8]. Another research found that 17% of older adults with diabetes experienced catastrophic healthcare expenditure on medications [9]. The availability and affordability of anti-diabetic medicines were pivotal for patients with diabetes. As a chronic, progressive condition, diabetes required a long-term drug therapy. A cost-effective treatment available to patients with diabetes may lead to substantial reductions in morbidity and mortality [10]. However, according to a report, only 25.8% of Chinese patients received diabetes treatment [11]. One important reason for no treatment was poor availability and lack of affordability of essential medicines for diabetes [12].

In China, patients can obtain prescription medicines from hospitals, primary care institutions, and private pharmacies with a prescription from a physician [13]. Although more than 95% of Chinese people were covered by social health insurance [14], mostly only inpatient expenditures can be refunded. In recent years, the health insurance started to consider outpatient reimbursement for chronic diseases. However, only essential medicines purchased from specific pharmacies (usually one patient can specify two public hospitals and one private pharmacy) can be reimbursed [15]. In China, majority of the essential medicines were generics.

Therefore, for patients with diabetes, lower availability of essential anti-diabetic generics, especially in public hospitals, meant higher out-of-pocket payment, which would make them unaffordable.

Several studies have examined the availability, price and affordability of medicines in China using the standardized method developed by World Health Organization (WHO) and Health Action International (HAI) [13, 16–19]. The results generated by these researches showed that generally the medicine price was decreasing with time but the availability was also decreasing. And the medicines' availability in public hospitals were lower than that in private pharmacies, especially generics. Among the previous studies, two focused on insulin products, and others evaluated the price and availability of general medicines which only included three oral anti-diabetic medicines (glibenclamide, gliclazide and metformin). Besides, the WHO/HAI survey manual recommended to survey only one specific strength and dosage for each medicine. As there were many strengths and dosages for each medicine in China, using a standardized WHO/HAI method would underestimate the availability of medicines.

We have previously investigated the availability and prices of medicines in Shaanxi Province of China in different years and also in Pakistan [18–23], including an investigation of prices, price component, availability and affordability of insulin products in Shaanxi province as supported by WHO [19]. The objective of this study was to assess the price, availability and affordability of oral anti-diabetic medicines in Shaanxi Province in western China. We conducted a cross sectional survey from May to July in 2015 in Shaanxi Province using a modified WHO/HAI methodology, which included all the strengths and dosages of each medicine. To the best of our knowledge, this was the first study of its type in western China focusing on oral anti-diabetic medicines. We believed it would be helpful to government policymakers, researchers and practitioners.

## Methods

### Study setting

Shaanxi was located in western China, with a population of 37.93 million and 11 areas in its jurisdiction, ranked 14th for GDP per capita in mainland China (31 provinces in total in the mainland) in 2015 [24]. Shaanxi was broadly representative of the typical health of the 12 western provinces of China, and in 2012 the Ministry of Health of China and WHO selected Shaanxi as one of the three pilot regions for the western area health initiative to explore key health issues in western China [25].

### Sampling

According to the WHO/HAI manual, Xi'an (the capital city of Shaanxi Province) was selected as the major urban center. An additional five areas (Yulin, Xianyang, Baoji, Shangluo and Weinan) reachable within one day's travel of Xi'an were randomly chosen.

In each survey area, we first selected the main public hospital (usually a tertiary hospital). Additional 11 public hospitals per survey area were then randomly selected three hour's drive from the main hospital. The public sector sample therefore contained 12 public hospitals (including 2 tertiary hospitals, 4 secondary hospitals and 6 primary hospitals) in each of the six survey areas, for a total of 72 public hospitals. The private sector sample was identified by selecting the private pharmacy closest to each of the selected public hospitals. In total, 72 public hospitals and 72 private pharmacies were included. Among the 72 public hospitals, 33 hospitals (all the tertiary hospitals and 21 secondary hospitals) had 15% profit margin for medicines, and 39 hospitals (all the primary hospitals and 3 secondary hospitals) had implemented zero mark-up policy.

## Medicine selection

We focused on oral anti-diabetic medicines and excluded insulin in our survey. The reasons were two folds. Firstly, insulin was not the first choice of most patients in China. Insulin is a life-saving medicine for people with type 1 diabetes and is used to manage an increasing number of people with type 2 diabetes [26]. However, according to the latest National Health Service survey in China, only 15% of patients with diabetes chose insulin for treatments [27]. Secondly, the information of insulin (availability and price) was provided in two papers in Shaanxi and Hubei Provinces in China [13, 19]. For most research which surveyed the price and availability, only included 2 or 3 oral anti-diabetic medicines. To the best of our knowledge, currently no research provided the information of availability and price of all the oral-diabetic medicines in Shaanxi.

There were 3 oral medicines for diabetes suggested by WHO/HAI manual, including glibenclamide from the global core list, metformin and gliclazide from the regional core list. In addition to these 3 medicines, we included 5 supplementary medicines, which were selected by referring to "China guideline for type 2 diabetes (2013)" [28] and "Standard Therapeutic Guidelines for National Essential Drugs" [29]. In total, 8 medicines were selected. Of these, 6 were essential medicines, and two were not.

We identified all the dosages and strengths for each medicine used in Shaanxi Province by referring to the information provided by the provincial Food and Drug Administration, which is the official institution responsible for administration of pharmaceuticals in Shaanxi province. Finally, 31 different dosages or strengths were included for the 8 medicines, covering originator brands (OBs) and generics (Table 1). Because the original brand of glibenclamide were not used in Shaanxi Province, we only included the generics.

**Table 1. Medicines selected for survey.**

| Name | Medicine type | Dosage forms and strengths |
|---|---|---|
| Acarbose* | OB | 50 mg tab |
| | Generics | 50 mg tab/cap |
| Glibenclamide* | Generics | 2.5 mg tab |
| Gliclazide# | OB | 30 mg SR tab, 80 mg tab |
| | Generics | 30 mg SR tab, 80 mg tab |
| Glimepiride* | OB | 2 mg tab |
| | Generics | 1 mg tab, 2 mg tab/cap |
| Glipizide* | OB | 5 mg CR tab |
| | Generics | 2.5 mg tab/cap, 5 mg tab/cap, 5 mg CR tab |
| Metformin* | OB | 500 mg tab, 800 mg tab |
| | Generics | 250 mg tab/cap, 500 mg tab, 250 mg SR cap, 500 mg SR tab, 250 mg and 500 mg R cap/tab |
| Pioglitazone | OB | 15 mg tab |
| | Generics | 5 mg tab, 15 mg tab/cap, 30 mg tab |
| Repaglinide | OB | 1 mg tab, 2 mg tab |
| | Generics | 0.5 mg tab, 1 mg tab, 2 mg tab |

SR: sustained release; CR: controlled release; R: retard. OB: originator brands.

*: National essential medicine

#: Provincial essential medicine

## Data collection and entry

Twelve well-trained data collectors were organized in pairs to visit medicine outlets and record medicine availability and price on the day of the survey, using a standardized data collecting form (S1 File). For each generic, as there were many manufacturers, we only collected the price of lowest-priced generic (LPG).

Two trained graduate students entered all the survey data into the pre-programmed Excel Workbook (WHO/HAI 2015) using a double entry technique.

## Statistical analysis

**1) Availability.**   The availability of each medicine was reported as the percentage of outlets in which the medicine was found on the day of data collection. If one outlet had at least a specific dosage or strength of one medicine, this medicine was denoted as available in this outlet.

The following ranges were used for describing availability: not available (availability = 0), very low (0< availability<30%), low (30%≤availability<50%), fairly high (50%≤availability< 80%), high (availability ≥80%) [30].

**2) Price.**   For one medicine, if the original brand and generic had the same dosage form and strength, we compared their prices using median price variation ratio. Besides, the median price ratio (MPR) was used for evaluation if the medicine had international reference price (IRP) in the Management Sciences for Health (MSH) 2015 Price Indicator Guide [31]. The median price was not calculated when the medicine was present in fewer than 3 outlets. The formulation of MPR were as follows.

$$\text{MPR} = \frac{\text{the median price of one medicine}}{\text{IRP}} \times 100\%$$

**3) Affordability.**   To assess affordability, the standard treatment for each medicine was included. If the standard treatment cost 1 day's wages or less, it was considered to be affordable. As rural and urban residents had different levels of income, we used different variables to calculate. For urban residents, we used the average daily wage of the lowest-paid unskilled government workers in Shaanxi Province, which was RMB 44.1667 (USD 7.2304) at the time of this survey [32]. For rural residents, we used the average daily net-income of rural residents in Shaanxi Province in 2015, which was RMB 21.8(USD 3.5688) [33]. The calculation process of standard treatment cost was:

The standard treatment cost = the median price per mg × daily dose × standard treatment duration
Where, the daily dose and the standard treatment duration were converted to mgs and days, respectively. If a medicine treatment cost less than 1 day's wages or net income, we regarded its affordability as good.

We tested normality and homogeneity of variances of prices and affordability, and performed either parametric (independent $t$ test or paired $t$ test) or non-parametric (Mann-Whitney U test) analyses to compare the differences of those indictors between public and private sectors. All statistical analyses were conducted using SPSS 18.0 and a $p$ value <0.05 was considered as statistical significance.

## Ethics

The Ethics Committee of Xi'an Jiaotong University Health Science Center (Xi'an, China) reviewed this study and stated that no formal ethics approval was required in this particular case. Oral consents were obtained from all participating organizations.

## Results

### Availability

The mean availability of OBs and generics was 34.3% and 28.7%, respectively in the public hospitals, and 44.1% and 64.4% in the private pharmacies (Table 2). Among all the hospitals, tertiary hospitals had highest availability of both OBs and generics, wherever primary hospitals had the lowest. And hospitals with 15% mark-up for medicines had higher availability of both OBs and generics than hospitals without. In both sectors, metformin and acarbose had the highest availability among all the OBs and generics, respectively. In general, the mean availability of sampled medicines was higher in the private pharmacies than the public hospitals except the OBs of repaglinide and pioglitazone. Most of the generic medicines were more available than originator brands in both sectors, except gliclazide, acarbose and repaglinide.

In the public hospitals, the availability of six generics were low (less than 50%), and both the OB and generics of repaglinide have fairly high availability (Table 3). In the private pharmacies, the availability of most generics were more than 50% (fairly high or high), and gliclazide, metformin and acarbose were highly available (more than 80%).

### Price

Comparing the price between OBs and LPGs, we found that the price of OB was 3.38 and 3.16 times the LPGs on average in public hospitals and private pharmacies (Table 4). In public hospitals, the biggest difference between OB and LPGs was Glipizide (OB/LPG = 6.47), and the smallest was Acarbose (OB/LPG = 1.44). While in private pharmacies, the biggest difference between OB and LPGs was metformin (OB/LPG = 6.58), and the smallest was repaglinide (2 mg tab, OB/LGP = 1.05). Overall, the OBs of acarbose, gliclazide (80 mg tab and 30 mg SR tab), glimepiride, metformin and repaglinide (1 mg and 2 mg tab) were higher priced in public hospitals than in private pharmacies ($p<0.05$). While for generics, except of acarbose which was higher priced in public hospitals, the other medicines were similarly priced in public and private sectors. However, taken the 9 standard treatment as a whole, the day's wages to pay for urban and rural residents, there was no significant difference between public and private sectors ($P>0.05$).

**Table 2. Availability of anti-diabetic medicines in public hospitals and private pharmacies in Shaanxi Province, China in 2015 (%).**

|  | Tertiary hospital | | Secondary hospital | | Primary hospital | | Hospitals with mark-up | | Hospitals without mark-up | | All public hospitals | | All private pharmacies | |
|---|---|---|---|---|---|---|---|---|---|---|---|---|---|---|
|  | OBs | Gs | OBs | Gs | OBs | Gs | OBs | Gs | OBs | Gs | OBs | Gs | OBs | Gs |
| Acarbose | 100.0 | 58.3 | 83.3 | 29.2 | 47.2 | 19.4 | 9.1 | 36.4 | 4.9 | 23.1 | 68.1 | 29.2 | 91.7 | 50.0 |
| Glibenclamide | / | 8.3 | / | 0 | 0 | 5.6 | / | 9.1 | / | 2.6 | / | 5.6 | / | 20.8 |
| Gliclazide | 83.3 | 25.0 | 58.3 | 45.8 | 25.0 | 22.2 | 66.7 | 36.4 | 23.1 | 25.6 | 45.8 | 30.6 | 83.3 | 75.0 |
| Glimepiride | 50.0 | 58.3 | 25.0 | 29.2 | 5.6 | 22.2 | 36.4 | 42.4 | 5.1 | 2.1 | 19.4 | 19.4 | 29.2 | 80.6 |
| Glipizide | 16.7 | 50.0 | 4.2 | 50.0 | 2.8 | 44.4 | 9.1 | 48.5 | 2.6 | 46.2 | 5.6 | 47.2 | 5.6 | 80.6 |
| Metformin | 44.4 | 91.7 | 58.3 | 100.0 | 25.0 | 91.7 | 69.7 | 97.0 | 20.5 | 92.3 | 44.4 | 94.4 | 81.9 | 97.2 |
| Pioglitazone | 16.7 | 33.3 | 0 | 0 | 0 | 0 | 6.1 | 21.2 | 0 | 0 | 2.8 | 29.2 | 0 | 31.9 |
| Repaglinide | 100.0 | 25.0 | 66.7 | 4.2 | 30.6 | 0 | 81.8 | 12.1 | 30.8 | 0 | 54.2 | 54.2 | 16.7 | 79.2 |
| Mean | 58.7 | 43.7 | 42.3 | 32.3 | 19.5 | 25.7 | 39.8 | 37.9 | 12.4 | 24.0 | 34.3 | 28.7 | 44.1 | 64.4 |

/: No data for this medicine, because the original brand of glibenclamide were not used in Shaanxi Province. OBs: originator brands. Gs: generics.

**Table 3. Availability rating of anti-diabetic medicines.**

| Availability | Public hospitals | | Private pharmacies | |
|---|---|---|---|---|
| | OBs | Gs | OBs | Gs |
| Not available | None | None | Pioglitazone | None |
| Very low | Glimepiride Glipizide Pioglitazone | Acarbose Glibenclamide Glimepiride Pioglitazone | Glipizide Glimepiride Repaglinide | Glibenclamide |
| Low | Gliclazide Metformin | Gliclazide Glipizide | None | Pioglitazone |
| fairly high | Acarbose Repaglinide | Repaglinide | None | Acarbose Gliclazide Repaglinide |
| High | None | Metformin | Acarbose Gliclazide Metformin | Glimepiride Glipizide Metformin |

OBs: originator brands. Gs: generics.

Among all these medicines, there were 3 medicines with 4 different dosage forms and strengths, which had IRPs. Comparing their median price with IRPs, the results showed that only the LGP of gliclazide (80 mg) had lower price than the IRP. The median patient price for the 4 OBs was 12.38 times the IRP in public hospitals, and 10.26 times in private pharmacies. For the 4 LPGs, the median patient price was 4.52 and 2.81 times the IRP in public hospitals and private private pharmacies, respectively. The public hospitals sold OB glimepiride at extremely high price, with 21.25 times the IRP (Table 5).

## Affordability

Table 6 and Table 7 showed the affordability of 9 standard treatments for diabetes in both sectors. All the OB treatments cost more than 1 day's wages except gliclazide (80 mg). If patients choose treatment with the LPGs, the medicines would be much more affordable. For example, for urban residents 1 month's treatment with glipizide controlled release (5 mg per day) purchased from public hospitals required 1.89 days' wages for the OB, but just 0.26 days' wages for its generic equivalent. The affordability of these medicines for urban residents was much better

**Table 4. The median price (RMB) of anti-diabetic medicines.**

| Medicines | | Public hospitals | | | Private pharmacies | | | Comparison between public and private* | |
|---|---|---|---|---|---|---|---|---|---|
| | | OBs | LPGs | OBs/LPGs | OBs | LPGs | OBs/LPGs | P for OB | P for LPGs |
| Acarbose | 50 mg tab | 2.47 | 1.71 | 1.44 | 1.77 | 1.40 | 1.26 | 0.000 | 0.000 |
| Glibenclamide | 2.5 mg tab | / | 0.02 | | / | 0.02 | | | |
| Gliclazide | 80 mg tab | 1.24 | 0.27 | 4.59 | 1.05 | 0.31 | 3.39 | 0.000 | 0.435 |
| Gliclazide | 30 mg SR tab | 1.89 | 0.53 | 3.57 | 1.60 | 0.65 | 2.46 | 0.042 | 0.612 |
| Glimepiride | 2 mg tab | 5.19 | 1.22 | 4.25 | 3.87 | 1.23 | 3.15 | 0.000 | 0.714 |
| Glipizide | 5 mg CR tab | 2.78 | 0.43 | 6.47 | 2.64 | 0.45 | 5.87 | 0.064 | |
| Metformin | 500 mg tab | 1.45 | 0.95 | 1.53 | 1.25 | 0.19 | 6.58 | 0.000 | 0.056 |
| Pioglitazone | 15 mg tab | / | / | | / | 5.05 | | | |
| Repaglinide | 1 mg tab | 2.47 | 1.35 | 1.83 | 1.93 | 1.29 | 1.50 | 0.000 | 0.651 |
| Repaglinide | 2 mg tab | 2.80 | / | | 2.27 | 2.17 | 1.05 | 0.000 | |
| Mean | | | | 3.38 | | | 3.16 | | |

/: No value, because when the medicine was present in fewer than 3 outlets, the median price was not calculated.

*: *t* test or ANOVA for data with normal distribution; Mann-Whitney U tests for data with non-normal distribution

**Table 5. MPRs of four anti-diabetic medicines with IRP.**

| Medicine | Public hospitals | | Private pharmacies | |
|---|---|---|---|---|
| | OBs | LPGs | OBs | LPGs |
| Gliclazide 80 mg tab | 4.16 | 0.90 | 3.53 | 1.04 |
| Gliclazide 30 mg SR tab | 9.50 | 2.64 | 9.06 | 3.26 |
| Glimepiride 2 mg tab | 21.25 | 5.00 | 15.83 | 5.03 |
| Metformin 500 mg tab | 14.60 | 9.55 | 12.63 | 1.89 |
| Mean | 12.38 | 4.52 | 10.26 | 2.81 |

for rural residents. All the LGPs in public hospitals were much more affordable than that in private pharmacies, except acarbose and repaglinide. On the contrary, all the OBs in private pharmacies were much more affordable than that in public hospitals, except repaglinide.

## Discussion

In the present study, we evaluated the availability, price and affordability of anti-diabetic medicines using a modified WHO/HAI methodology in Shaanxi Province, China. There were two main findings: 1) Three anti-diabetic OBs and 3 generics met the WHO's availability target (80% availability)[34] in private sector, and only 1 generics met this target in public sector; 2) OBs cost much more than their generic equivalents, and treatment with OBs was unaffordable, especially for rural residents.

There was enormous availability difference between public hospitals and private pharmacies. OBs had higher availability than generics in public hospitals, while opposite situation was observed in the private pharmacies. Because in China, previously the government allowed the public hospitals a 15% profit margin on drugs, which induced serious health hazards and physicians tended to over-prescribe, especially expensive medicines [24]. Even after 2017, this 15% mark-up was cancelled for all the public hospitals, physician in public hospitals still had higher financial incentives to prescribe expensive OBs as they can get more grey income from pharmaceutical companies [35]. And another possible reason could contribute to this result was that most physicians in China believed OBs had better clinical results than generics. The

**Table 6. Costs (RMB) of standard treatments with original brand anti-diabetic medicines.**

| Originators | Dosage per day (mg) | Median price per mg | | Cost for a duration | | Day's wages to pay for treatment for urban residents | | | Day's income to pay for treatment for rural residents | | |
|---|---|---|---|---|---|---|---|---|---|---|---|
| | | Public | Private | Public | Private | Public | Private | P* | Public | Private | P* |
| Acarbose | 150 | 0.049 | 0.035 | 220.5 | 157.5 | 4.99 | 3.57 | 0.09 | 10.11 | 7.22 | 0.09 |
| Glibenclamide | 5 | / | / | / | / | / | / | | / | / | |
| Gliclazide (30 mg) | 30 | 0.063 | 0.0533 | 56.7 | 47.97 | 1.28 | 1.09 | | 2.60 | 2.20 | |
| Gliclazide (80 mg) | 80 | 0.016 | 0.0131 | 38.4 | 31.44 | 0.87 | 0.71 | | 1.76 | 1.44 | |
| Glimepiride | 2 | 2.597 | 1.9334 | 155.82 | 116.004 | 3.53 | 2.63 | | 7.15 | 5.32 | |
| Glipizide | 5 | 0.556 | 0.5286 | 83.4 | 79.29 | 1.89 | 1.80 | | 3.83 | 3.64 | |
| Metformin | 1000 | 0.003 | 0.002 | 90 | 60 | 2.04 | 1.36 | | 4.13 | 2.75 | |
| Pioglitazone | 30 | / | / | / | / | / | / | | / | / | |
| Repaglinide | 2 | 1.428 | 1.667 | 85.68 | 100.02 | 1.94 | 2.26 | | 3.93 | 4.59 | |
| Mean | | 0.67 | 0.61 | 104.36 | 84.60 | 2.36 | 1.92 | | 4.79 | 3.88 | |

/: No value, the original brand medicines were not available.

*: paired *t* tests.

**Table 7. Costs (RMB) of standard treatments with anti-diabetic LPGs.**

| Generics | Dosage per day (mg) | Median price per mg | | Cost for a duration | | Day's wages to pay for treatment for urban residents | | | Day's income to pay for treatment for rural residents | | |
|---|---|---|---|---|---|---|---|---|---|---|---|
| | | Public | Private | Public | Private | Public | Private | P* | Public | Private | P* |
| Acarbose | 150 | 0.034 | 0.028 | 153 | 126.0 | 3.46 | 2.85 | 0.81 | 7.02 | 5.78 | 0.72 |
| Glibenclamide | 5 | 0.007 | 0.008 | 1.05 | 1.2 | 0.02 | 0.03 | | 0.05 | 0.06 | |
| Gliclazide (30 mg) | 30 | 0.017 | 0.021 | 15.3 | 18.9 | 0.35 | 0.43 | | 0.70 | 0.87 | |
| Gliclazide (80 mg) | 80 | 0.003 | 0.004 | 7.2 | 9.6 | 0.16 | 0.22 | | 0.33 | 0.44 | |
| Glimepiride | 2 | 0.610 | 0.625 | 36.6 | 37.5 | 0.83 | 0.85 | | 1.38 | 1.72 | |
| Glipizide | 5 | 0.086 | 0.090 | 12.9 | 13.5 | 0.26 | 0.31 | | 0.59 | 0.62 | |
| Metformin | 1000 | 0.0005 | 0.001 | 15.0 | 30.0 | 0.34 | 0.68 | | 0.69 | 1.38 | |
| Pioglitazone | 30 | 0.146 | 0.162 | 131.4 | 145.8 | 2.98 | 3.30 | | 6.03 | 6.69 | |
| Repaglinide | 2 | 1.380 | 1.327 | 82.8 | 79.62 | 1.87 | 1.80 | | 3.80 | 3.65 | |
| Mean | | 0.25 | 0.25 | 50.56 | 51.35 | 1.15 | 1.16 | | 2.32 | 2.36 | |

*: paired *t* tests.

Chinese government has not required generic drugs to have the same quality and efficacy as the original drugs until the early of 2016. Therefore before 2016, local generics were not bio-equivalent with the originators and were deemed of lower quality [36]. While in private pharmacies, they were more willing to provide lowest-priced generics to gain more customers because of fierce market competition. In general, the availability in the private pharmacies was better than in the public hospitals. This could be the consequence of regulation issued by Chinese Ministry of Health in 2006, which required all public health institutions should purchase one medicine with no more than two types of dosage forms for injection and oral medicines [37]. The difference of availability was particularly true in our survey as all the strengths and dosage forms were accounted. Using this methodology, the availability of medicines increased a lot comparing to previous studies using standard WHO/HAI method. For example, the availability of gliclazide generics evaluated by Jiang et al [23] was only 45% in private pharmacies, while in our study it increased to 75%.

Among all these medicines, the availability of glibenclamide was especially low. The OB of glibenclamide was not available, and the availability of its generics was very low in both sectors. This was similar with the result of Guan et al. [38]. As we talked with several clinical pharmacists, they told us that currently the overall use of sulfonylureas was relatively low as the sulfonylurea agents had a higher risk of hypoglycemic reactions comparing with other oral hypoglycemic medicines, and they hardly used glibenclamide, as it was a long-acting sulfonylurea agent which had more adverse effects.

For OBs, we found that the price was higher in public hospitals than in private pharmacies. But for generics, there was no significant difference between different sectors. We speculated that this finding occurred because most OBs had higher availability at hospitals with 15% mark-up than hospitals without mark-ups. In addition, intensified competition also made the private pharmacies charge less.

The affordability of OBs was poor. Although the affordability of LPGs was better, more than half of these medicines were not affordable for rural residents. Especially for acarbose, repaglinide and pioglitazone, even the urban residents cannot afford the treatment with these LPGs. But even worse, the availability of these LPGs in the public sector was poor, and the majority of patients must purchase medicine from the private pharmacies or buy more expensive brand-name drugs. Moreover, under current medical insurance scheme, mostly patients

had to pay their own expenses if they bought medicines from private pharmacies. These factors can increase the economic burden for patients.

Compared with insulin products, the oral anti-diabetic medicines had lower availability but higher affordability. Li et al.'s [19] reported the availability and affordability of insulin in Shaanxi province, all three kinds of insulin products (prandial, basal and premixed insulin) had a 100% availability in public hospitals, and a fairly high availability in private sector ranging from 62.5% to 68.8%, while in our survey even the highest availability (generics in private pharmacy) was only 64.4%. For the affordability, the insulin products would cost 3.5 to 17.1 days' wage of lowest-paid government worker (urban residents) in Shaanxi, while even the oral anti-diabetic OBs just cost 0.71 to 4.99 days' wage for urban residents. Another study about insulin in Hubei province [16] generated similar results as Li et al. The big price gap between insulin products and oral anti-diabetics maybe was one important reason why there was only 15% of patients with diabetes choosing insulin for treatments

Compared with general medicines, the availability of oral anti-diabetic in our survey was higher. Fang et al.[18] survey 50 medicines in Shaanxi in 2012, the availability of the 8 oral anti-diabetic OBs were 3.94 and 2.77 times of the availability of OBs of the 50 general medicines in public and private sectors, respectively; 1.4 and 1.82 times for generics in public and private sectors.

This study had some limitations. First, the availability of anti-diabetic medicines was measured at specific facilities on the day of the survey. The facilities surveyed may normally have a product in stock, but they may have run out of the medicine on the day of data collection. We may not accurately capture the availability of medicines. Secondly, our study was limited to one province only. Therefore the results may not be generalizable to the whole country. Thirdly, the affordability was calculated for single medicine for diabetes, whereas patients may take multi-medicine at a time. So, the affordability may be overestimated.

## Conclusion

The availability of most generics was fairly high in private pharmacies, but low in public hospitals. In both sectors most generics were more available than OBs. The high prices of OBs had severely influenced the affordability of medicines, especially for the rural residents. An effective policy should be initiated to ensure patients had better access to more affordable anti-diabetic medicines.

## Supporting information

**S1 File. Data collecting form.**
(DOCX)

## Acknowledgments

We appreciate the cooperation and participation of the pharmacists and other staff at the medicine outlets where data was collected.

## Author Contributions

**Data curation:** Caijun Yang, Shuchen Hu, Wenwen Zhu.

**Formal analysis:** Shuchen Hu, Wenwen Zhu.

**Funding acquisition:** Caijun Yang.

**Investigation:** Caijun Yang, Wenwen Zhu.

**Methodology:** Caijun Yang, Zongjie Li.

**Project administration:** Yanbing Zhu, Yu Fang.

**Resources:** Yanbing Zhu.

**Software:** Wenwen Zhu.

**Supervision:** Yu Fang.

**Validation:** Wenwen Zhu, Zongjie Li.

**Writing – original draft:** Caijun Yang, Shuchen Hu.

**Writing – review & editing:** Caijun Yang.

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
