## [Decision Letter · Decision Letter 0]

24 Jul 2019

PONE-D-19-15076

Evaluating access to anti-diabetic medicines: A cross-sectional survey of prices, availability and affordability in Shaanxi Province, western China

PLOS ONE

Dear Dr. Fang,

Thank you for submitting your manuscript to PLOS ONE. After careful consideration, we feel that it has merit but does not fully meet PLOS ONE’s publication criteria as it currently stands. Therefore, we invite you to submit a revised version of the manuscript that addresses the points raised during the review process.

We would appreciate receiving your revised manuscript by Sep 07 2019 11:59PM. To enhance the reproducibility of your results, we recommend that if applicable you deposit your laboratory protocols in protocols.io, where a protocol can be assigned its own identifier (DOI) such that it can be cited independently in the future. For instructions see: http://journals.plos.org/plosone/s/submission-guidelines#loc-laboratory-protocols

We look forward to receiving your revised manuscript.

Kind regards,

Lutz Heide

Academic Editor

PLOS ONE

Additional Editor Comments (if provided):

In addition to the points raised by the reviewers, please add a sentence or paragraph in the introduction section citing ALL your previous publications on medicine availablity, prices and affordability in China, in Pakistan (and possibly in further countries), e.g.: "We have previously investigated the availablity and prices of medicines in different parts of China and in other countries (citations), including an investigation of prices, availability and affordability of insulin products in Shaanxi province (citation)."

I would have preferred if you had combined your previous data on insulin (Trop Med Int Health. 2019), and your current data on other antidiabetic medicines (both from Shaanxi province), into a single publication. I advise you to combine such data in future, in order to avoid the impression that you attempt to make as many publications as possible out of a limited amount of data.

Journal Requirements:

2) Please include additional information regarding the standardized data collecting form used in the study and ensure that you have provided sufficient details that others could replicate the analyses. For instance, if you developed a data collecting form as part of this study and it is not under a copyright more restrictive than CC-BY, please include a copy, in both the original language and English, as Supporting Information.

Reviewers' comments:

Reviewer's Responses to Questions

**Comments to the Author**

1. Is the manuscript technically sound, and do the data support the conclusions?

Reviewer #1: Yes

Reviewer #2: Yes

2. Has the statistical analysis been performed appropriately and rigorously? 

Reviewer #1: Yes

Reviewer #2: No

3. Have the authors made all data underlying the findings in their manuscript fully available?

Reviewer #1: Yes

Reviewer #2: Yes

4. Is the manuscript presented in an intelligible fashion and written in standard English?

Reviewer #1: Yes

Reviewer #2: Yes

5. Review Comments to the Author

Reviewer #1: The study assesses availability and prices of oral anti-diabetic medicines in Shaanxi province in Western China, using a slightly modified WHO/HAI methodology to measure availability and prices of essential medicines.

It is a sound research, that includes a large basket of survey points (public hospitals and private pharmacies). It is based on an established methodology whose further development is justified. The presented evidence on availability and prices of medicines in different parts of the world is appreciated and needed.

The paper is well written in standard English and meets the requirements of a scientific study. I congratulate the author on this article.

I see one major limitation:

1. The study did not include insulin. This is only briefly mentioned in the limitations but no reason is provided. It is reported that treatment with insulin was “effective and safe” (in China or the studied province?) but no evidence (reference) is provided to justify this statement. I would like to see an explicit mentioning of the exclusion of insulin from the study in the methods section, including an explanation why authors decided to do so.

In order to improve the quality of the article, I recommend considering the following suggestions:

2. As the article is intended for an international readership, some background information on the Chinese health and pharmaceutical system is missing. This would allow the readers putting the findings into context. There is some information provided in the discussion, e.g.

- financial incentives (lines 239 ff.)

- the issue with the mark-ups, mentioned in brackets (lines 257ff.)

- patients purchasing out-of-pocket medicines and the reference to the medical insurance scheme (lines 264ff.)

It would be appreciated if these pieces of information were nicely presented in the introduction. Also, it would be helpful for readers to understand the supply/dispensing channels. For instance, can outpatients get prescriptions from (public) hospitals?

3. I would like to see the findings of this study discussed in the light of current literature, in particular against the backdrop of other WHO/HAI studies performed on China (even if they were on other medicines) as well as of other price studies on diabetes medicines. I do not consider references 25 and 26 sufficient. In this respect, it is a pity that insulins were excluded from the study since reference studies on diabetes medication, such as the ACCISS (Addressing the Challenges and Constraints of Insulin Sources and Supply) project undertaken by HAI, usually refer to insulins.

4. The study was performed in one province of China. This should also be stated explicitly in line 107 (as it is done in the abstract). It is totally fine to limit the scope of the study to one province. However, it would be appreciated to add some background information on this province, including an assessment whether, or not, this province is representative for China.

5. Throughout the article, the terminology “lowest-priced generic” / LPG, based on the WHO/HAI methodology, is used. In the WHO/HAI context, this is correct since the methodology only includes a specific dosage and strength of a medicine. However, the authors further developed the methodology to include all dosages and strengths. As such, I would not talk of LPGs in the section on availability but simply of generics. The term LPG works, however, well for the sections on the prices.

6. There are some inconsistencies in the text, sometimes OBs are mentioned first followed by generics, and in other parts (see also the order of Tables 6 and 7) it is the other way round. This should be harmonised.

7. More out of curiosity: I see that the price survey was performed in Q2/2015. Why was the paper submitted only in 2019?

Further specific comments:

8. Lines 74-75: “Although an additional diagnostic criterion was included after 2010” → this is not fully clear, kindly rephrase/explain

9. Lines 82-83: “its complications” → this is not fully clear, kindly rephrase

10. Line 102: “the WHO/HAI survey manual only surveyed one …” → it is suggested to rephrase into “the WHO/HAI survey manual recommends to survey only one …” (since the manual does not survey)

11. Line 131: “we found” → “we identified”?

12. Line 132: What is the role of the provincial Food and Drug Administration? Is it the marketing authorisation authority (as the FDA in the US)? Does marketing authorisation differ between the provinces of China?

13. Line 144: “using a standardized data collecting form” → kindly mention that this is the form provided by the WHO/HAI manual (if you used this form), and could you provide an English translated version of the form (or a summary) in the Supplementary Materials

14. Line 179: “participated” → “participating”

15. Line 199: “/: No data” – there is no missing data in the table

16. Line 209, line 227 and line 228: RMB – this should probably read “MPR”?

17. Consistencies in having a blank and not having a blank between a figure and “mg”

18. Line 228: first row of Table 7 should be “originator”, not generic

19. Lines 233 and 278: “the availability of anti-diabetic medicines was not optimistic”: I strongly suggest rephrasing. In this respect, it would be good to set the information in the context to some baseline indicator, such as WHO’s 80% availability of affordable essential medicines, including generics, to treat major non-communicable diseases (NCDs), in the public and private sectors of countries by 2025 (see also: Ewen M, Zweekhorst M, Regeer B, Laing R. Baseline assessment of WHO’s target for both availability and affordability of essential medicines to treat non-communicable diseases. PLOS ONE. 2017;12(2):e0171284).

Reviewer #2: This is a normal evaluation study on medicine access. Basically it was conducted based on WHO/HAI methodology and adjusted according to China's national conditions. The findings are relatively reliable. However, still some minor revisions should be made.

Introduction

1.Line 96-97 Please explained the reasons why many patients do not receive diabetes treatmen in more detail.

2.Line 100-102 The literature analyses could be updated. A research artical published in 2018 included 20 antidiabetic drugs in Huibei, China. So the statement “this was the first study of its type in China focusing on oral anti-diabetic medicines” in Line 109 is not exactly accurate.

Results

3.The sector or drug category comparisons should be based on some statistical tests, includuing t test and etal, especially for Table 4-7.

Discussion

4.Line 238 The preference for OBs due to financial incentives could be an import factor, but it should not be the only one. This finding should be discussed more.

5.As for this speculation in 256-258, some analysis can be added in the results section, such as in this survey how many OBs were availability at hospitals with 15% mark-up than hospitals

6.Line 265-266 The availability of LPGs in the public sector was poor, and the majority of patients must purchase medicine from the private pharmacies or buy more expensive brand-name drugs, which could increase the economic burden for patients. However, the Table 4 and 5 showed the median price of anti-diabetic medicines in private pharmacies was lower than public hospitals.

7.Line 267 The China's new health insurance policy for chronic diseases such as diabetes could be more conducive to improve affordability.

6. PLOS authors have the option to publish the peer review history of their article (what does this mean?). If published, this will include your full peer review and any attached files.

Reviewer #1: Yes: Sabine Vogler

Reviewer #2: No

---

## [Author Response · Author response to Decision Letter 0]

4 Sep 2019

Response to Reviewers

Reviewer #1

The study assesses availability and prices of oral anti-diabetic medicines in Shaanxi province in Western China, using a slightly modified WHO/HAI methodology to measure availability and prices of essential medicines.

It is a sound research that includes a large basket of survey points (public hospitals and private pharmacies). It is based on an established methodology whose further development is justified. The presented evidence on availability and prices of medicines in different parts of the world is appreciated and needed.

The paper is well written in standard English and meets the requirements of a scientific study. I congratulate the author on this article.

Response: Thanks for your valuable comments!

I see one major limitation:

1. The study did not include insulin. This is only briefly mentioned in the limitations but no reason is provided. It is reported that treatment with insulin was “effective and safe” (in China or the studied province?) but no evidence (reference) is provided to justify this statement. I would like to see an explicit mentioning of the exclusion of insulin from the study in the methods section, including an explanation why authors decided to do so.

Response: Thanks for your comments! We added the reason for choosing the oral medicines in “Medicine Selection” in part of “Methods” like following:

“We focused on oral anti-diabetic medicines and excluded insulin in our survey. The reasons were two folds. Firstly, insulin was not the first choice of most patients in China. Insulin is a life-saving medicine for people with type 1 diabetes and is used to manage an increasing number of people with type 2 diabetes [26]. However, according to the latest National Health Service survey in China, only 15% of patients with diabetes chose insulin for treatments [27]. Secondly, the information of insulin (availability and price) was provided in two papers in Shaanxi and Hubei Provinces in China [13, 19]. For most research which surveyed the price and availability, only included 2 or 3 oral anti-diabetic medicines. To the best of our knowledge, currently no research provided the information of availability and price of all the oral-diabetic medicines in Shaanxi.”

In order to improve the quality of the article, I recommend considering the following suggestions:

2. As the article is intended for an international readership, some background information on the Chinese health and pharmaceutical system is missing. This would allow the readers putting the findings into context. There is some information provided in the discussion, e.g.

- financial incentives (lines 239 ff.)

- the issue with the mark-ups, mentioned in brackets (lines 257ff.)

- patients purchasing out-of-pocket medicines and the reference to the medical insurance scheme (lines 264ff.)

It would be appreciated if these pieces of information were nicely presented in the introduction. Also, it would be helpful for readers to understand the supply/dispensing channels. For instance, can outpatients get prescriptions from (public) hospitals?

Response: Thanks for your comments! We rewrote the introduction and added more information about the patients purchasing medicines and the medical insurance scheme like following:

“In China, patients can obtain prescription medicines from hospitals, primary care institutions, and private pharmacies with a prescription from a physician [13]. Although more than 95% of Chinese people were covered by social health insurance [14], mostly only inpatient expenditures can be refunded. In recent years, the health insurance started to consider outpatient reimbursement for chronic diseases. However, only essential medicines purchased from specific pharmacies (usually one patient can specify two public hospitals and one private pharmacy) can be reimbursed [15]. In China, majority of the essential medicines were generics. Therefore, for patients with diabetes, lower availability of essential anti-diabetic generics, especially in public hospitals, meant higher out-of-pocket payment, which would make them unaffordable. ”

To make it naturally, we added the financial incentives and issue with mark-ups in the discussion part as following:

“There was enormous availability difference between public hospitals and private pharmacies. OBs had higher availability than generics in public hospitals, while opposite situation was observed in the private pharmacies. Because in China, previously the government allowed the public hospitals a 15% profit margin on drugs, which induced serious health hazards and physicians tended to over-prescribe, especially expensive medicines [24]. Even after 2017, this 15% mark-up was cancelled for all the public hospitals, physician in public hospitals still had higher financial incentives to prescribe expensive OBs as they can get more grey income from pharmaceutical companies [35]. And another possible reason could contribute to this result was …….”

3. I would like to see the findings of this study discussed in the light of current literature, in particular against the backdrop of other WHO/HAI studies performed on China (even if they were on other medicines) as well as of other price studies on diabetes medicines. I do not consider references 25 and 26 sufficient. In this respect, it is a pity that insulins were excluded from the study since reference studies on diabetes medication, such as the ACCISS (Addressing the Challenges and Constraints of Insulin Sources and Supply) project undertaken by HAI, usually refer to insulins.

Response: Thanks for your comments! We rewrote the discussion and compared our results with previous studies as following:

“Compared with insulin products, the oral anti-diabetic medicines had lower availability but higher affordability. Li et al.’s [19] reported the availability and affordability of insulin in Shaanxi province, all three kinds of insulin products (prandial, basal and premixed insulin) had a 100% availability in public hospitals, and a fairly high availability in private sector ranging from 62.5% to 68.8%, while in our survey even the highest availability (generics in private pharmacy) was only 64.4%. For the affordability, the insulin products would cost 3.5 to 17.1 days’ wage of lowest-paid government worker (urban residents) in Shaanxi, while even the oral anti-diabetic OBs just cost 0.71 to 4.99 days’ wage for urban residents. Another study about insulin in Hubei province [16] generated similar results as Li et al. The big price gap between insulin products and oral anti-diabetics maybe was one important reason why there was only 15% of patients with diabetes choosing insulin for treatments.

Compared with general medicines, the availability of oral anti-diabetic in our survey was higher. Fang et al.[18] survey 50 medicines in Shaanxi in 2012, the availability of the 8 oral anti-diabetic OBs were 3.94 and 2.77 times of the availability of OBs of the 50 general medicines in public and private sectors, respectively; 1.4 and 1.82 times for generics in public and private sectors. ”

4. The study was performed in one province of China. This should also be stated explicitly in line 107 (as it is done in the abstract). It is totally fine to limit the scope of the study to one province. However, it would be appreciated to add some background information on this province, including an assessment whether, or not, this province is representative for China.

Response: Thanks for your comments! We modified according to your suggestion in the end of the background and also in the methods part as following: 

In line 107, we changed to “The objective of this study was to assess the price, availability and affordability of oral anti-diabetic medicines in Shaanxi Province in western China.” (Line 121 in the new version)

In methods part, we added “study setting”, and introduced the Shaanxi briefly as:

“Shaanxi was located in western China, with a population of 37.93 million and 11 areas in its jurisdiction, ranked 14th for GDP per capita in mainland China (31 provinces in total in the mainland) in 2015 [24]. Shaanxi was broadly representative of the typical health of the 12 western provinces of China, and in 2012 the Ministry of Health of China and WHO selected Shaanxi as one of the three pilot regions for the western area health initiative to explore key health issues in western China [25]..”

5. Throughout the article, the terminology “lowest-priced generic” / LPG, based on the WHO/HAI methodology, is used. In the WHO/HAI context, this is correct since the methodology only includes a specific dosage and strength of a medicine. However, the authors further developed the methodology to include all dosages and strengths. As such, I would not talk of LPGs in the section on availability but simply of generics. The term LPG works, however, well for the sections on the prices.

Response: Thanks for your comments! We changed the expression of LPG to generics in availability (including abstract, results and discussion parts). 

6. There are some inconsistencies in the text, sometimes OBs are mentioned first followed by generics, and in other parts (see also the order of Tables 6 and 7) it is the other way round. This should be harmonised.

Response: Thanks for your comments! We went through the paper and changed the inconsistencies according to your suggestion.

7. More out of curiosity: I see that the price survey was performed in Q2/2015. Why was the paper submitted only in 2019?

Response: Thanks for your question! We submitted this paper to one journal in the early of 2017 which rejected us in the beginning of 2018 because they did not find any reviewer; then we submitted to another journal, after 6 months without no response, we withdrew it and then submitted to Plos One. 

Further specific comments:

8. Lines 74-75: “Although an additional diagnostic criterion was included after 2010” → this is not fully clear, kindly rephrase/explain

Response: Thanks for your comments! We explained it in the new version like this: “Although an additional diagnostic criterion HbA1c (glycated hemoglobinA1c, concentration of 6.5% or higher) was included after 2010”. 

9. Lines 82-83: “its complications” → this is not fully clear, kindly rephrase

Response: Thanks for your comments! We explained it in the new version: 

“Health expenditure related to diabetes and its complications (such as periodontal disease, vision loss, diabetic foot, end-stage renal disease and cardiovascular disease, etc.)”

10. Line 102: “the WHO/HAI survey manual only surveyed one …” → it is suggested to rephrase into “the WHO/HAI survey manual recommends to survey only one …” (since the manual does not survey)

Response: Thanks for your comments, we modified the paper according to your suggestion. 

11. Line 131: “we found” → “we identified”?

Response: Thanks for your comments, we changed the expression according to your suggestion. 

12. Line 132: What is the role of the provincial Food and Drug Administration? Is it the marketing authorisation authority (as the FDA in the US)? Does marketing authorisation differ between the provinces of China?

Response: Similar with FDA in the USA, the State Food and Drug Administration in China (SFDA) is responsible for the supervision and administration of registration of pharmaceuticals, medical devices, and to develop national standards for pharmaceuticals, medical devices. The provincial Food and Drug Administration is responsible for administration of pharmaceuticals in province. The responsibility of Food and Drug Administration in different provinces is the same. 

We added an explanation following the appearance of provincial Food and Drug Administration as following: “We identified all the dosages and strengths for each medicine used in Shaanxi Province by referring to the information provided by the provincial Food and Drug Administration ,which is the official institution responsible for administration of pharmaceuticals in Shaanxi province”

13. Line 144: “using a standardized data collecting form” → kindly mention that this is the form provided by the WHO/HAI manual (if you used this form), and could you provide an English translated version of the form (or a summary) in the Supplementary Materials.

Response: Thanks for your suggestion! We uploaded an English version of data collecting form in the supplementary materials as suggested. 

14. Line 179: “participated” → “participating”

Response: Thanks for your comments, we corrected the error.

15. Line 199: “/: No data” – there is no missing data in the table

Response: Thanks for your comments! We are sorry for our mistake, and corrected the error in the new version.

16. Line 209, line 227 and line 228: RMB – this should probably read “MPR”?

Response: Thanks for your comments! In the three places, there should be RMB, it is the official currency of China. Here it meant that the median price in RMB of each drug. To make it more understandable, we also mentioned the exchange rate of USD to RMB in the affordability part of “Statistical Analysis”. 

17. Consistencies in having a blank and not having a blank between a figure and “mg”

Response: Thanks for your comments! We went through the paper and made it consistent according to your suggestion. 

18. Line 228: first row of Table 7 should be “originator”, not generic

Response: Thanks for your comments! We are sorry for our mistake, and corrected the error in the new version.

19. Lines 233 and 278: “the availability of anti-diabetic medicines was not optimistic”: I strongly suggest rephrasing. In this respect, it would be good to set the information in the context to some baseline indicator, such as WHO’s 80% availability of affordable essential medicines, including generics, to treat major non-communicable diseases (NCDs), in the public and private sectors of countries by 2025 (see also: Ewen M, Zweekhorst M, Regeer B, Laing R. Baseline assessment of WHO’s target for both availability and affordability of essential medicines to treat non-communicable diseases. PLOS ONE. 2017;12(2):e0171284).

Response: Thanks for your comments! We referred this paper, and changed the expression to “There were two main findings: 1) Three anti-diabetic OBs and 3 generics met the WHO’s availability target (80% availability)[34] in private sector, and only 1 generics met this target in public sector”. 

Reviewer #2

This is a normal evaluation study on medicine access. Basically it was conducted based on WHO/HAI methodology and adjusted according to China's national conditions. The findings are relatively reliable. However, still some minor revisions should be made.

Introduction

1.Line 96-97 Please explained the reasons why many patients do not receive diabetes treatment in more detail.

Response: Thanks for your comments. For patients do not receive diabetes treatments, we think there are two most important reasons: 1) not aware of their condition (as reported by one article published in JAMA, only 30.1% were aware of their condition. 2) poor availability and lack of affordability of anti-diabetic medicines. As in this paper we only focused on medicines, we modified the expression in the article. 

2. Line 100-102 The literature analyses could be updated. A research article published in 2018 included 20 antidiabetic drugs in Huibei, China. So the statement “this was the first study of its type in China focusing on oral anti-diabetic medicines” in Line 109 is not exactly accurate.

Response: Thanks for your comments. We searched Pubmed and did not found such kind of paper in international journals. But in local journals, we found several in other provinces. Therefore, we updated the literature and to make it accurate, we changed the statement to “To the best of our knowledge, this was the first study of its type in western China focusing on oral anti-diabetic medicines.”

Results

3.The sector or drug category comparisons should be based on some statistical tests, including t test and etal, especially for Table 4-7.

Response: Thanks for your comments! We added statistical test in Table 4 for each OB and LPG, comparing the price between public hospitals and private pharmacies. As for Table 5, there were median price rations of only four medicines, we thought it would not be that suitable for statistical tests. Table 6-7 showed the affordability of standard treatments for diabetes, we performed paired t test (each medicine in public and private sectors) for the 9 medicines. 

Discussion

4.Line 238 The preference for OBs due to financial incentives could be an import factor, but it should not be the only one. This finding should be discussed more.

Response: Thanks for your comments. There are many reasons for physicians’ preference for OBs. In the new version, we modified this part and provided another one important reason as following: 

“There was enormous availability difference between public hospitals and private pharmacies. OBs had higher availability than generics in public hospitals, while opposite situation was observed in the private pharmacies. Because in China, previously the government allowed the public hospitals a 15% profit margin on drugs, which induced serious health hazards and physicians tended to over-prescribe, especially expensive medicines [24]. Even after 2017, this 15% mark-up was cancelled for all the public hospitals, physician in public hospitals still had higher financial incentives to prescribe expensive OBs as they can get more grey income from pharmaceutical companies [35]. And another possible reason could contribute to this result was that most physicians in China believed OBs had better clinical results than generics. The Chinese government has not required generic drugs to have the same quality and efficacy as the original drugs until the early of 2016. Therefore before 2016, local generics were not bioequivalent with the originators and were deemed of lower quality [36].”

5.As for this speculation in 256-258, some analysis can be added in the results section, such as in this survey how many OBs were availability at hospitals with 15% mark-up than hospitals

Response: Thanks for your comments! We added this results in Table 2 columns “Hospitals with mark-up” and “Hospitals without mark-up”. The results showed that hospitals with 15% mark-up for medicines had higher availability of both OBs and generics than hospitals without 15% mark-up.

6. Line 265-266 The availability of LPGs in the public sector was poor, and the majority of patients must purchase medicine from the private pharmacies or buy more expensive brand-name drugs, which could increase the economic burden for patients. However, the Table 4 and 5 showed the median price of anti-diabetic medicines in private pharmacies was lower than public hospitals.

Response: Thanks for your comments. As under current medical insurance scheme, mostly patients had to pay their own expenses if they bought medicines from private pharmacies. That is why we said purchasing from private pharmacies could increase the economic burden for patients. We added some background information in the introduction and also added a sentence for explanation in this paragraph in the new version to make it easier to understand. 

In the Background: “Although more than 95% of Chinese people were covered by social health insurance [14], mostly only inpatient expenditures can be refunded. In recent years, the health insurance started to consider outpatient reimbursement for chronic diseases. However, only essential medicines purchased from specific pharmacies (usually one patient can specify two public hospitals and one private pharmacy) can be reimbursed [15]. In China, majority of the essential medicines were generics. Therefore, for patients with diabetes, lower availability of essential anti-diabetic generics, especially in public hospitals, meant higher out-of-pocket payment, which would make them unaffordable.”

In the discussion: “The affordability of OBs was poor. Although the affordability of LPGs was better, more than half of these medicines were not affordable for rural residents. Especially for acarbose, repaglinide and pioglitazone, even the urban residents cannot afford the treatment with these LPGs. But even worse, the availability of these LPGs in the public sector was poor, and the majority of patients must purchase medicine from the private pharmacies or buy more expensive brand-name drugs. Moreover, under current medical insurance scheme, mostly patients had to pay their own expenses if they bought medicines from private pharmacies. These factors can increase the economic burden for patients.”

7.Line 267 The China's new health insurance policy for chronic diseases such as diabetes could be more conducive to improve affordability.

Response: Thanks for your valuable comments! We added some information about health insurance policy for chronic diseases in part of “Introduction”. And we modified our expression in discussion. 

Additional Editor Comments:

In addition to the points raised by the reviewers, please add a sentence or paragraph in the introduction section citing ALL your previous publications on medicine availablity, prices and affordability in China, in Pakistan (and possibly in further countries), e.g.: "We have previously investigated the availablity and prices of medicines in different parts of China and in other countries (citations), including an investigation of prices, availability and affordability of insulin products in Shaanxi province (citation)."

I would have preferred if you had combined your previous data on insulin (Trop Med Int Health. 2019), and your current data on other antidiabetic medicines (both from Shaanxi province), into a single publication. I advise you to combine such data in future, in order to avoid the impression that you attempt to make as many publications as possible out of a limited amount of data.

Response: Thanks for your valuable comments and suggestion! We added our previous publications on this topic in the introduction as “We have previously investigated the availability and prices of medicines in Shaanxi Province of China in different years and also in Pakistan [18-23], including an investigation of prices, price component, availability and affordability of insulin products in Shaanxi province as supported by WHO [19]. The objective of this study”.

Thanks for you great suggestion! We will consider this in the future. Because the survey about insulin and the oral-diabetic medicines were implemented by different students, they belong to different projects. The insulin survey was organized by WHO, included small number of hospitals and pharmacies, and also focused on the price component part. The survey for oral-diabetic medicines only considered price and availability, but more outlets were included. We also compared the results of oral-diabetic medicines and insulin in discussion part.

---

## [Decision Letter · Decision Letter 1]

30 Sep 2019

Evaluating access to oral anti-diabetic medicines: A cross-sectional survey of prices, availability and affordability in Shaanxi Province, western China

PONE-D-19-15076R1

Dear Dr. Fang,

We are pleased to inform you that your manuscript has been judged scientifically suitable for publication and will be formally accepted for publication once it complies with all outstanding technical requirements.

With kind regards,

Lutz Heide

Academic Editor

PLOS ONE

Additional Editor Comments (optional):

Please see Review Comments to the Authors, and correct accordingly.

Reviewers' comments:

Reviewer's Responses to Questions

**Comments to the Author**

1. If the authors have adequately addressed your comments raised in a previous round of review and you feel that this manuscript is now acceptable for publication, you may indicate that here to bypass the “Comments to the Author” section, enter your conflict of interest statement in the “Confidential to Editor” section, and submit your "Accept" recommendation.

Reviewer #1: All comments have been addressed

2. Is the manuscript technically sound, and do the data support the conclusions?

Reviewer #1: Yes

3. Has the statistical analysis been performed appropriately and rigorously? 

Reviewer #1: Yes

4. Have the authors made all data underlying the findings in their manuscript fully available?

Reviewer #1: Yes

5. Is the manuscript presented in an intelligible fashion and written in standard English?

Reviewer #1: Yes

6. Review Comments to the Author

Reviewer #1: The authors have well responded to my questions and have addressed my comments appropriately. I suggest publication of the article.

The only comment that I still have is to have a final edit in terms of language and syntax, in particular for the revised parts, e.g.

Line 144: I suggest writing “is” instead of “was” and end the sentence of “western China”.s

Lines 282/283: remove the line break in the sentence

Line 301: full stop is missing at the end of the sentence

7. PLOS authors have the option to publish the peer review history of their article (what does this mean?). If published, this will include your full peer review and any attached files.

Reviewer #1: Yes: Sabine Vogler

---

## [Editor Report · Acceptance letter]

7 Oct 2019

PONE-D-19-15076R1 

Evaluating access to oral anti-diabetic medicines: A cross-sectional survey of prices, availability and affordability in Shaanxi Province, western China 

Dear Dr. Fang:

I am pleased to inform you that your manuscript has been deemed suitable for publication in PLOS ONE. Congratulations! Your manuscript is now with our production department. 

With kind regards,

on behalf of

Dr. Lutz Heide 

Academic Editor

PLOS ONE